# Green Hierarchical Vision Transformer for Masked Image Modeling

Lang Huang[1], Shan You[2]*, Mingkai Zheng[3], Fei Wang[2], Chen Qian[2], Toshihiko Yamasaki[1]

[1]The University of Tokyo; [2]SenseTime Research; [3]The University of Sydney

{langhuang,yamasaki}@cvm.t.u-tokyo.ac.jp

{youshan,wangfei,qianchen}@sensetime.com

## Abstract

We present an efficient approach for Masked Image Modeling (MIM) with hierarchical Vision Transformers (ViTs), allowing the hierarchical ViTs to discard masked patches and operate only on the visible ones. Our approach consists of three key designs. First, for window attention, we propose a Group Window Attention scheme following the Divide-and-Conquer strategy. To mitigate the quadratic complexity of the self-attention w.r.t. the number of patches, group attention encourages a uniform partition that visible patches within each local window of arbitrary size can be grouped with equal size, where masked self-attention is then performed within each group. Second, we further improve the grouping strategy via the Dynamic Programming algorithm to minimize the overall computation cost of the attention on the grouped patches. Third, as for the convolution layers, we convert them to the Sparse Convolution [25, 13] that works seamlessly with the sparse data, *i.e.*, the visible patches in MIM. As a result, MIM can now work on most, if not all, hierarchical ViTs in a green and efficient way. For example, we can train the hierarchical ViTs, *e.g.*, Swin Transformer [49] and Twins Transformer [14], about $2.7\times$ faster and reduce the GPU memory usage by 70%, while still enjoying competitive performance on ImageNet classification and the superiority on downstream COCO object detection benchmarks.[†]

## 1 Introduction

Driven by the great success of Masked Language Modeling (MLM) [56, 57, 15, 6] in natural language processing (NLP) and the advancement of Vision Transformers (ViTs) [17, 49, 66, 77], Masked Image Modeling (MIM) emerged as a promising self-supervised pre-training paradigm for computer vision (CV). MIM learns representations from unlabelled data by masked prediction, *e.g.*, predicting the discrete tokens [3], the latent features [81, 68, 2], or the raw pixels [28, 73] of the randomly masked input image patches. Among them, the representative work Masked Autoencoder (MAE) [28] exhibited competitive performance as well as impressive efficiency. In essence, MAE proposed an asymmetric encoder-decoder architecture for MIM, where the encoder (*e.g.*, a standard ViT model [17]) operates only on visible patches, and the light-weight decoder recovers all patches for mask prediction.

On the one hand, the asymmetric encoder-decoder architecture significantly reduces the computation burden of pre-training. On the other hand, MAE only supports the isotropic ViT [17] architecture as the encoder, while most of the modern vision models adopt hierarchical structure [43, 31, 49], in part due to the need of handling the scale-variations of visual elements. In fact, the hierarchical structure and local inductive bias are crucial in various CV tasks that require representations of different levels

---

*Corresponding author.    [†]Code and pre-trained models: https://github.com/LayneH/GreenMIM.

36th Conference on Neural Information Processing Systems (NeurIPS 2022).

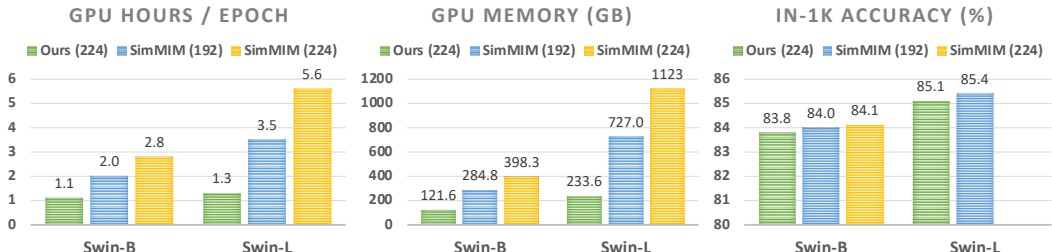

Figure 1: **Comparison with SimMIM in terms of efficiency.** All methods use a Swin-B/Swin-L backbone and batch size of 2,048. The experiments of our method are conducted on a single machine with eight V100 GPUs, CUDA 10, and PyTorch 1.8, while those of SimMIM require 2 or 4 machines.

or scales to make predictions, including image classification [31], and object detection [22]. Yet it is still not straightforward how the hierarchical vision transformers, *e.g.* Swin Transformer [49], can be integrated into the MAE framework. Moreover, though the work SimMIM [73] has explored Swin Transformer for MIM, it operates on both visible and masked patches and suffers from heavy computation costs compared with MAE. As a concrete example, we find that even the base size model of SimMIM cannot be trained on a single machine with eight 32GB GPUs, let alone the larger ones. The computation burden makes it difficult for a wider range of researchers to dive into this field of research, not even to mention the amount of carbon emission during the model development process.

To this end, we strive to devise a new and green approach for MIM with hierarchical models, in the spirit of Green AI [61, 74]. Our work focuses on extending the asymmetric encoder-decoder architecture of MAE to hierarchical vision transformers, particularly the representative model Swin Transformer [49], for the sake of efficient pre-training on visible patches only. We identify that the major obstacle is the inductive bias of hierarchical ViTs, i.e., the locality induced by i) window attention [36, 49, 77] with non-overlapped window partition, and ii) convolution/pooling [66, 19, 14, 27] with overlapped window partition. These operators are incompatible with random masking as it creates various-sized local windows that are infeasible for computing in parallel.

This paper provides the first attempt to address this drawback. We present a Green Hierarchical Vision Transformer for Masked Image Modeling that advocates a more practical method with drastically improved efficiency. Our methodology is conceptually simple and consists of three key designs.

1. Guided by the Divide-and-Conquer principle, we present a Group Window Attention scheme first partitioning the local windows with uneven numbers of visible patches into several equal-sized groups and then applying the masked attention within each group.

2. We formulate the aforementioned group partition as a constrained optimization problem, where the objective is to find a group partition that minimizes the computation cost of the attention on the grouped tokens. Inspired by the concept of Dynamic Programming [5] and the greedy principle, we propose an Optimal Grouping algorithm that adaptively selects the optimal group size and partitions the local windows into a minimum number of groups.

3. We convert the convolution layers in the hierarchical ViTs to the Sparse Convolution [25, 13, 1], which was originally designed for handling sparse point cloud data and works seamlessly with the masked inputs in MIM.

Our methodology is generic and does not make *any* modification to the architecture of the backbone models, such that we can make apple-to-apple comparisons with the baseline operating on both visible and masked patches. In our experimental evaluations, we observed that our method requires substantially less training time and consumes much less GPU memory while performing on par with the baseline on both ImageNet1K [60] classification and MS-COCO [47] object detection. Concretely, using Twins-L [14]/Swin-B [49]/Swin-L [49], our method achieves up to 2.7× speedup and consumes as few as 30% of GPU memory compared with the baseline SimMIM at the pre-training stage, while achieving 83.9%/83.8%/85.1% top-1 fine-tuning accuracy on the ImageNet-1K that is on par with those of SimMIM.

## 2 Related Works

**Self-Supervised Learning.**  Representation learning is a long-standing and fundamental question in CV. For a long time, representation learning had been dominated by supervised learning. Until recent three years, self-supervised learning (SSL) exhibited impressive performance and attained significant attention. Generally, SSL solves a proxy task without the actual interest to learn good representations. According to the proxy tasks, SSL methods can be categorized into generative approaches and discriminative approaches. Generative approaches predict the original data based on the partially observed inputs [65, 55, 44], predict the transformation applied to the input [52, 21], or model pixels in the input space [42, 23, 39, 33]. Masked Image Modeling also falls into this category. Discriminative approaches received more interest during the past few years, especially the contrastive learning methods. Contrastive learning creates multiple views of images with a set of random data distortions and encourages the representations to be invariant to the distortions. A large number of contrastive learning approaches [70, 53, 29, 10, 79, 80] drive the training by maximizing the similarities between positive samples (*i.e.*, views from the same image) while minimizing those between the negative samples, and some works simply get rid of the negative pairs [26, 7, 11, 37, 78, 8]. Beyond contrastive learning on global features, several methods proposed to maintain the spatial information of representations and use region/mask/pixel-level contrastive learning [67, 72, 71, 32, 35].

**Masked Language/Image Modeling.**  Self-supervised pre-training has revolutionized the NLP. Among them, the Masked Language Modeling (MLM) proposed in BERT [15] and its variants [6] are the most dominant methods, which learn representations by predicting the tokens that are randomly masked in the input sentence. Masked Image Modeling has a similar idea of predicting corrupted images, and some of these methods [65, 55] were proposed Even preceded to BERT. These methods were, however, unable to perform on par with other pre-training paradigms at that time. Until recently, aided by the significant advancement of Vision Transformers [17], several MIM methods presented promising results [17, 3, 28, 73, 68] and became the state-of-the-art of self-supervised learning in CV. These methods can be roughly differentiated in terms of the prediction target, *e.g.*, color bins [17], discrete tokens [3, 16] from pre-trained VAEs [63, 58], raw pixels [28, 73], and handcrafted features [68]. Among these approaches, MAE [28] exhibited competitive performance as well as impressive efficiency as it discards the masked tokens and operates only on the visible ones.

**Isotropic and Hierarchical Vision Transformers.**  The seminal work Vision Transformer (ViT) [17] revolutionized the conventional view of images. ViT, and its variant [62], treat an image as a sequence of patches and adopt a pure Transformer [64] backbone to model the patches for image classification, achieving an impressive performance even compared with the Convolutional Neural Networks. Nevertheless, while the results of ViT are promising in classification, its performance on dense prediction tasks is less favorable, which is largely due to its low-resolution feature maps inherited from its isotropic structure and the quadratic complexity of self-attention [64]. To this end, a strand of works proposed hierarchical structure [66, 49, 19, 77], efficient attention [36, 49, 14, 38], and locality bias [18, 69, 27, 76] for ViTs, unleashing the potential of ViTs as general-purpose vision backbones. Our work performs studies upon the representative hierarchical ViTs, Swin Transformer [49] and Twins Transformer [14] but generalizes to any other ViTs with window local attention or convolution/pooling.

**Green AI.**  Witnessing the exponential growth of computations of big AI models [15, 6, 48], the concept of Green AI attains mounting attention in recent years [61, 74]. Rather than being merely obsessed with accuracy, Green AI advocates making efficiency an important measure of AI models, championing the greener approaches that are more inclusive to the research community. This work follows the path of Green AI and presents a greener approach for MIM with hierarchical ViTs.

## 3 Approach

### 3.1 Preliminary

**Notations.**  Let $\mathbf{X} \in \mathbb{R}^{C \times H \times W}$ denote the input feature where $C$, $H$, and $W$ are the numbers of channels, height, and width of $\mathbf{X}$; $\mathbf{M} \in \{0,1\}^{H \times W}$ denotes the (spatial) mask generated randomly during training where $0$ indicates a patch is invisible for the encoder, and vice versa.

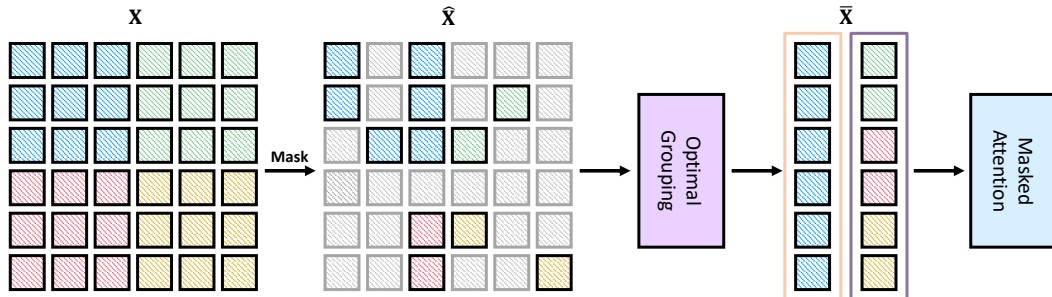

Figure 2: **Illustration of the Group Window Attention scheme.** In Masked Image Modeling (MIM), the input $\mathbf{X}$, where different colors indicate the tokens belong to different local windows, is randomly masked, producing $\widehat{\mathbf{X}}$ of which most tokens are invisible. Our Group Window Attention first performs an optimal grouping to group the visible tokens of the local windows into several equal-sized groups, forming $\overline{\mathbf{X}}$. Finally, we perform the Masked Attention within each group to ensure no inter-window information leakage.

**Masked Image Modeling.** MIM learns representations by predicting the masked portion of a input $\mathbf{X}$ from its partial observation $\widehat{\mathbf{X}} \leftarrow \mathrm{Mask}(\mathbf{X}, \mathbf{M})$. Existing MIM methods fall into two categories regarding the $\mathrm{Mask}(\cdot, \cdot)$ operation. Most methods [3, 73, 68] use the Hadamard product for masking and retain the masked patches, *i.e.*, $\widehat{\mathbf{X}} \leftarrow \mathrm{Mask}(\mathbf{X}, \mathbf{M}) = \mathbf{X} \odot \mathbf{M}$ with $\mathbf{M}$ broadcasted for $C$ times along the channel dimension. In sharp contrast to these methods, Masked Autoencoders (MAE) [28] proposes throwing the masked patches at the masking stage, *i.e.*,

$$\widehat{\mathbf{X}} \leftarrow \mathrm{Mask}(\mathbf{X}, \mathbf{M}) = \{\mathbf{X}_{i,j} : \mathbf{M}_{i,j} = 1\}. \tag{1}$$

MAE designs an asymmetric and isotropic encoder-decoder architecture to take advantage of partial inputs: the encoder operates only on the visible patches $\widehat{\mathbf{X}}$ without mask tokens; the decoder reconstructs the original images from representations of visible patches and masked tokens. This design allows MAE to achieve competitive performance as well as impressive efficiency, *e.g.*, $3\times$ training speedup compared with the ones operating on all patches. Nevertheless, the MAE works only with the isotropic ViTs and it is unclear *how to translate the efficiency of MAE to hierarchical ViTs*, which exhibited nearly unanimous superiority over the isotropic ones on most vision tasks [66, 49, 19, 14, 77]. In this paper, we attempt to answer this question and propose a much greener approach for MIM with Hierarchical ViTs.

### 3.2 Green Hierarchical Vision Transformer for Masked Image Modeling

**Adapting existing hierarchical ViTs for MIM.** A typical hierarchical ViT mainly consists of Feed-Forward Networks (FFNs) and efficient attentions, which fail to operate only on the visible patches. We identify that the major obstacle is the inductive bias of hierarchical ViTs, i.e., the locality induced by i) window attention [36, 49, 77] with non-overlapped window partition, and ii) convolution/pooling [66, 19, 14, 27] in FFNs or attentions with overlapped window partition. These operators are incompatible with random masking as it creates various-sized local windows that are infeasible for computing in parallel. To this end, we present two key insights to make most, if not all, hierarchical ViTs be able to operate only on visible patches in a green manner.

**Group Window Attention.** Regarding the non-overlapped window partition, we propose a Group Window Attention scheme that significantly improves the computation efficiency of window attention on masked features. Given the masked feature $\widehat{\mathbf{X}} = \mathrm{Mask}(\mathbf{X}, \mathbf{M})$ following Equation (1), we collect a set of uneven local windows $\widehat{\mathbf{X}} \rightarrow \{\widehat{\mathbf{X}}_i\}_{i=0}^{n_w}$ of which each element contains the visible tokens only, with sizes $\{w_i\}_{i=0}^{n_w}$ accordingly. As shown in Figure 2, our Group Window Attention first uses an Optimal Grouping algorithm to partition the uneven windows into several equal-sized groups and then performs Masked Attention within each group to avoid information leakage. In the next two subsections, we will elaborate on these two components, respectively.

**Incorporating with Sparse Convolution.** Our intuition is the features of mask inputs can be viewed as sparse tensors where only features of (a small number of) visible patches are retained while

---

**Algorithm 1** Optimal Grouping

---

**Require:** The number of visible patches within each local window $\{w_i\}_{i=0}^{n_w}$,
 1: Minimum computational cost $c^* \leftarrow +\infty$
 2: **for** $g_s = \max_i \{w_i\}_{i=1}^{n_w}$ **to** $\sum_{i=1}^{n_w} w_i$ **do**
 3:     Remaining windows $\Phi \leftarrow \{w_i\}_{i=1}^{n_w}$; partition $\Pi \leftarrow \emptyset$; the number of group $n_g \leftarrow 0$
 4:     **repeat**
 5:         $\pi_{n_g} \leftarrow \text{Knapsack}(g_s, \Phi)$, as in Equation (7)
 6:         $\Pi \leftarrow \Pi \cup \pi_{n_g}$; $\Phi \leftarrow \Phi \setminus \pi_{n_g}$
 7:         $n_g \leftarrow n_g + 1$
 8:     **until** $\Phi = \emptyset$
 9:     $c \leftarrow \mathcal{C}(g_s, \Pi)$, as in Equation (8)
10:     **if** $c < c^*$ **then**
11:         $c^* \leftarrow c$; $\Pi^* \leftarrow \Pi$
12:     **end if**
13: **end for**
14: **return** Optimal group partition $\Pi^*$

---

the others are omitted. From this perspective, we propose to directly replace all the convolution/pooling operations in hierarchical ViTs with the highly optimized Sparse Convolution [24, 25, 13, 1], originally designed for the sparse 3D point cloud data.

### 3.3 Optimal Grouping with Dynamic Programming

**General formulation.** The first step of the optimal grouping is to find an indexes partition $\Pi$ with respect to the group size $g_s$:

$$\Pi = \{\pi_j\}_{j=1}^{n_g}, \tag{2}$$

$$s.t. \ \cup_j \ \pi_j = \{1, \cdots, \sum_i w_i\}, \ \sum_j |\pi_j| = \sum_i w_i, \text{and} \ \forall_j \sum_{k \in \pi_j} w_k \leq g_s. \tag{3}$$

where $n_g$ is the number of resulting groups. The conditions in Equation (3) constrain the partition to contain all local windows with no duplicate and enforce the actual size of each group to be smaller than $g_s$. Based on the partition $\Pi$, we obtain a set of grouped tokens $\{\bar{\mathbf{X}}_j\}_{j=1}^{n_g}$ that

$$\bar{\mathbf{X}}_j = \text{Concat}(\{\widehat{\mathbf{X}}_k : \forall k \in \pi_j\}), \ \bar{\mathbf{X}}_j \in \mathbb{R}^{C \times g_s}, \text{[2]} \tag{4}$$

upon which the Masked Attention is performed. Finally, we apply the inverse operation of the partition $\Pi$ to recover the positions of output tokens.

With the formulation above, there remain two questions unresolved: 1) how to choose the optimal group size $g_s^*$, and 2) how to obtain the optimal partition $\Pi^*$ given $g_s^*$. To this end, we formulate our objectives as the below min-min optimization problem,

$$g_s^* = \text{argmin}_{g_s} \ \mathcal{C}(g_s, \text{Partition}(g_s, \{w_i\}_i)), \tag{5}$$

$$\text{Partition}(g_s, \{w_i\}_{n_w}) = \text{argmin}_\Pi |\Pi|, \ s.t. \text{ Equation (3)}, \tag{6}$$

where $\mathcal{C}(\cdot)$ is a cost function measuring the computation cost of the attention with the grouped tokens. Intuitively, Equation (5) aims to find the optimal group size $g_s^*$ that the computation cost of the optimal partition w.r.t. $g_s^*$ is the minimum. Equation (6) searches for the optimal partition, with the constraints of Equation (3). Having the optimal group size, we can directly obtain the optimal partition $\Pi^* = \text{Partition}(g_s^*, \{w_i\}_{n_w})$. Next, we will present in detail how we solve the above optimization problem.

**Group partition with Dynamic Programming.** We find that the optimization problem in Equations (6) and (3) is a special case of the multiple subset sum problem with identical capacities (MSSP-I), a variant of the well-known 0-1 multiple knapsack problem with identical capacities (MKP-I) [40, Chapter 10]. In our case, the group size is analogous to the capacity of knapsacks, the

---

[2]Here we assume the number of tokens $\sum_i w_i$ is divisible by $g_s$ for simplicity. In practice, we pad the group $\pi_j$ when $|\pi_j|$ is smaller than $g_s$.

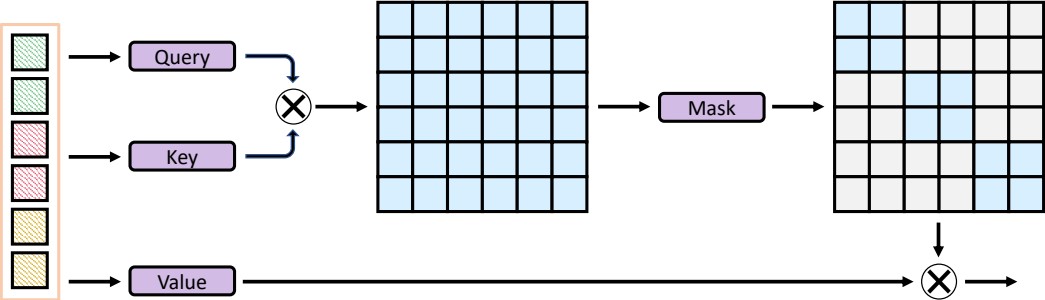

Figure 3: **Illustration of the Masked Attention scheme.** Given a group of tokens, we first compute their pairwise attention weights and then set the attention weights between tokens from different local windows to $-\infty$ (indicated by the gray cells). The final attention output is then computed with the masked attention map.

numbers of visible tokens $\{w_i\}_{n_w}$ are analogous to the values of goods, the weights of goods are the same as their values, and the number of knapsacks is unbounded. Although MSSP-I is strictly NP-complete, there exist multiple polynomial time approximation schemes for it, *e.g.*, using the dynamic programming (DP) algorithms [5]. Specifically, we make use of the DP algorithm for the single knapsack problem (or the subset sum problem):

$$\pi \leftarrow \text{Knapsack}(g_s, \Phi), \tag{7}$$

which selects a subset $\pi$ that $\sum_{u \in \pi} |u| \leq g_s$ out of the full set $\Phi$ (the pseudo-code of this algorithm is given in Appendix). We alternatively apply this algorithm to the remaining full set $\Phi$ and exclude the selected subset $\pi$ from $\Phi$, until $\Phi$ is empty. In practice, we found that our algorithm is very fast and the time cost is negligible because the number of local windows is often small, *e.g.*, less than 100 in our pre-training stage.

**Cost function.** Because we mainly care about the efficiency, we use the FLOPs to measure the computation complexity of the multi-head attention on grouped tokens, *i.e.*,

$$\mathcal{C}(g_s, \Pi) = |\Pi| \times (4g_s C^2 + 2g_s^2 C) = n_g \times (4g_s C^2 + 2g_s^2 C), \tag{8}$$

where $C$ is the number of channels. Although the complexity is quadratic w.r.t. the group size $g_s$, using smaller $g_s$ might produce more groups (and more padding) and suffers from suboptimal efficiency. Therefore the optimal group size is determined adaptively during training.

**Putting everything together.** We sweep over the possible values of group size, from $\max_i w_i$ to $\sum_i w_i$, to find the optimal group size. For each selected group size, we firstly use the DP algorithm in Equation (7) to partition the windows and then calculate the computation cost of the attention this partition. The one with minimum cost is chosen as the optimal group size. The pseudo-code of the optimal grouping is summarized in Algorithm 1.

### 3.4 Masked Attention

Because non-adjacent local windows are partitioned into the same groups, masking the attention weights is needed to avoid the information exchange between these local windows. As illustrated in Figure 3, having computed the attention map, we retain only the intra-window attention weights (*i.e.*, the block-diagonal elements) and discard the inter-window ones. A similar masking scheme is also applied to the retrieving of relative position bias [49], where we store the original absolute position of each token and compute the relative positions on-the-fly to retrieve the corresponding biases.

### 3.5 Batch-wise Random Masking

We observed that the per-sample random masking strategy would deteriorate the efficiency of our method: 1) it might produce different numbers of groups of local windows for each sample, which is intractable for the parallel computation of the Masked Attention; 2) when the mask patch size is smaller than the largest patch size of the hierarchical models, some patches might contain both visible

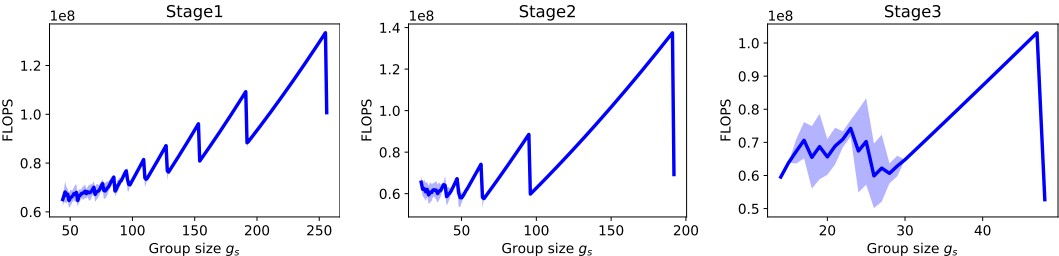

Figure 4: **The optimal group size $g_s$ at each stage.** The figure of the fourth stage is omitted here because there is only one local window in this stage, so the grouping is not necessary. The simulation is repeated 100 times, of which the mean and standard deviation (the shaded regions) are reported.

and masked inputs. In this case, we can not discard such patches during training and fail to fully take advantage of the sparsification. Therefore, we propose to set the mask patch size to the same value as the largest patch size of the encoder (*e.g.*, 32 for most hierarchical models, which is also the default choice of [73]) and use the same random mask for all samples in the same GPU device (a.k.a., a micro-batch).

# 4 Experiments

## 4.1 Implementation Details and Experimental Setups

We conduct experiments on the ImageNet-1K [60] (BSD 3-Clause License) image classification dataset and MS-COCO [47] (CC BY 4.0 License) object detection/instance segmentation dataset. The Swin-Base, Swin-Large [49] and Twins-Large [14] models, which consist of four stages with features of stride 4/8/16/32, are used as the encoder throughout this paper, allowing for direct comparisons with the baseline SimMIM [73] with Swin Transformer. The models are first pre-trained on the ImageNet-1K dataset without label and then fine-tuned on downstream tasks. All the experiments of our method are performed on a single machine with eight 32G Tesla V100 GPUs, CUDA 10.1, PyTorch [54] 1.8, and automatic mixed-precision training [51]. For the experiments involving convolution, we replace all the standard convolutions with the sparse convolution implemented in [1] as discussed in Sec. 3.2.

**Pre-training setup.** We patchify images of size $224\times224$ with a patch size of $4\times4$ and randomly mask the patches with a ratio $r$ ($r = 0.75$ by default) following the scheme in Section 3.5. The input images are first transformed by a set of simple data augmentations, including random cropping and horizontal flip, and standardization. Following the prior work MAE [28], we adopt a lightweight decoder that consists of $n_d$ (by default $n_d = 1$) standard transformer blocks with an embedding dimension of 512. The decoder takes the representations of visible patches and the mask token as input and is appended after the final stage of the encoder to learn representations for the masked patches. It is followed by a linear layer to predict the normalized pixel values of the masked patches. The models are trained for 100/200/400/800 epochs with a total batch size of 2,048. We use the AdamW optimizer [41] with the cosine annealing schedule [50]. We set the base learning rate to $1.5e^{-4}$, the weight decay to 0.05, the hyper-parameters of Adam $\beta_1 = 0.9$, $\beta_2 = 0.999$, the number of warmup epochs to 40 with an initial base learning rate $1.5e^{-7}$. The effective learning rate is scaled linearly by $\mathrm{batch\_size}/256$.

**Fine-tuning on the ImageNet-1K dataset.** For fine-tuning, we drop the decoder and directly append a 1,000-way fully-connected layer to the average-pooled output of the encoder as the classifier. The models are also optimized by the AdamW optimizer [41] with 100 training epochs in total, 20 warmup epochs, a base/warmup learning rate of $1.25e^{-4}/2.5e^{-7}$, the cosine annealing schedule [50], a weight decay of 0.05. A layerwise learning rate decay [3] of 0.9/0.8/0.9, and a stochastic depth [34] ratio of 0.2/0.3/0.2 are used for Swin-B/Swin-L/Twins-L respectively. The data augmentations are the same as [3, 73].

**Fine-tuning on the MS-COCO dataset.** We adopt the Mask R-CNN [30] architecture with the FPN [46] as the detector. All models are fine-tuned on the MS-COCO [47] 2017 train split (~118k images) and finally evaluated on the val split (~5k images). We use a batch size of 16, AdamW

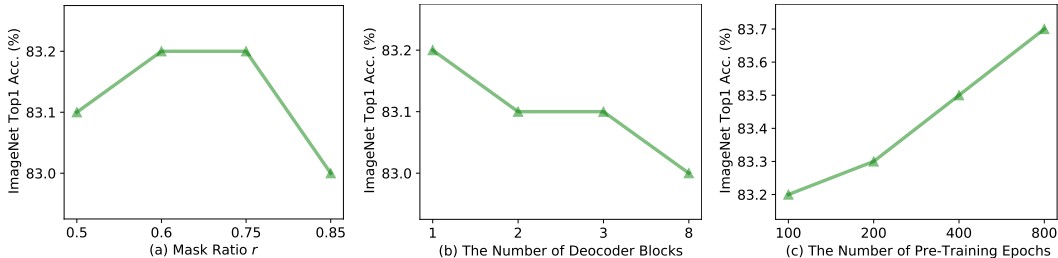

Figure 5: **Ablation studies of** (a) the choice of the mask ratio $r$, (b) the number of transformer blocks $n_d$ in the decoder, and (c) the number of pre-training epochs.

optimizer [41] with a learning rate of $1e^{-4}$, a weight decay of 0.05. The $1\times/3\times$ schedule in the mmdetection [9] is adopted, which uses 12/36 training epochs in total and decays the learning rate at the $\frac{3}{4}$ and $\frac{11}{12}$ of the total epochs by a factor of 10. The standard COCO metrics, including AP, $AP_{50}$, and $AP_{75}$ for both object detection and instance segmentation are used for evaluation.

## 4.2 Ablations studies

**Efficiency comparison with SimMIM.** We compare the efficiency of our method to the baseline SimMIM in Figure 1. The evaluations are performed on a single machine with eight 32GB V100 GPUs for our method and on 2 or 4 machines for the SimMIM because it fails to fit into a single machine with the default batch size of the original paper [73] (*i.e.*, 2,048). As we can see from the figures, the training of SimMIM with images of size $224^2$ is very slow and memory-hungry. Although training with smaller images considerably reduces the training time and memory consumption, it still far lags behind our method with images of size $224^2$. Concretely, with the same amount of training epochs, our method performs on par with baseline with ~$2\times$ speedup and ~60% of memory reduction using Swin-B. We also observe that the efficiency improvements become larger with the larger Swin-L, *e.g.*, $2.7\times$ speedup compared with $SimMIM_{192}$, highlighting the efficiency of our method with larger models. In addition, the efficiency comparison using Twins Transformer is presented in Appendix A.1 for completeness, which exhibits similar superiority of our approach.

**The optimal group size $g_s$ on each stage.** Because hierarchical models have multiple stages with different scales of features, the optimal group size of each stage might also be different. Regarding this, we design a simulation experiment to analyze the optimal $g_s$ at different stages. In the simulation, we randomly generate 100 masks following Section 3.5, compute the costs w.r.t. different choices of $g_s$, and report the mean/standard deviation of the costs in Figure 4. Note that the analysis of the 4th stage is omitted here because it only has one local window. In general, we observed that the cost increases quadratically w.r.t. the group size, except for some cases where the group size is exactly equal to the sum of a subset of local windows. Another intriguing observation is that the cost at each stage seems to be the minimum around the point that $g_s = 49$, which is equal to the window size of the window attention. This observation indicates that we may not need to sweep over all possible group sizes but simply set $g_s = p \times p$ in practice. Moreover, we analyze the influence of the optimal grouping and the cost of each part of it in Appendix A.2, showing that optimal grouping is indeed crucial for the efficiency of our method.

**The influence of the mask ratio, the number of decoder blocks, and the pre-training epochs.** From Figure 5(a), we can see that the performance of our method is quite stable with the mask ratio $r$ varying from 0.5 to 0.85, which conforms with the observation of [28]. In Figure 5(b), we also study the influence of the depth of the decoder. Intriguingly, the results suggest that fewer decoder blocks produce better results. This study favors the simple prediction head design of SimMIM [73] with hierarchical models and is in contrast to the observation of MAE [28] with isotropic ones. For simplicity and efficiency, we fix $r = 0.75$ and the number of decoder blocks to 1 throughout the paper. Furthermore, we study the impact of pre-training budgets on our method. As shown in Figure 5(c), the fine-tuning accuracy increases steadily w.r.t. the number of training epochs and does not seem to stagnate, suggesting its potential for a further performance boost.

**Pre-training with larger window size.** The work of [48] puts forth that using a larger window size is beneficial for fine-tuning. In practice, however, it might be less practical because of the

Table 2: Top1 accuracy on the ImageNet-1K validation set with the Swin-B or ViT-B models. All methods are trained with images of size $224 \times 224$ in both the pre-training and fine-tuning except for SimMIM$_{192}$ using $192 \times 192$ in the pre-training.

| Method | Model | #Params | PT Ep. | Ep. Hours | Total Hours | FT Ep. | Acc. (%) |
|---|---|---|---|---|---|---|---|
| *Training from scratch* | | | | | | | |
| Scratch, DeiT [62] | ViT-B | 86M | 0 | - | - | 300 | 81.8 |
| Scratch, MAE [28] | ViT-B | 86M | 0 | - | - | 300 | 82.3 |
| Scratch, Swin [49] | Swin-B | 88M | 0 | - | - | 300 | 83.5 |
| Scratch, Twins [14] | Twins-L | 99M | 0 | - | - | 300 | 83.7 |
| *Supervised Pre-training* | | | | | | | |
| Supervised, SimMIM [73] | Swin-B | 88M | 300 | - | - | 100 | 83.3 |
| Supervised, SimMIM [73] | Swin-L | 197M | 300 | - | - | 100 | 83.5 |
| *Pre-training with Contrastive Learning* | | | | | | | |
| MoCov3 [12] | ViT-B | 86M | 800 | - | - | 100 | 83.2 |
| DINO [8] | ViT-B | 86M | 800 | - | - | 100 | 82.8 |
| *Pre-training with Masked Image Modeling* | | | | | | | |
| BEiT [3] | ViT-B | 86M | 800 | - | - | 100 | 83.2 |
| MaskFeat [68] | ViT-B | 86M | 800 | - | - | 100 | 84.0 |
| MAE [28] | ViT-B | 86M | 1600 | 1.3 | 2069 | 100 | 83.6 |
| SimMIM$_{224}$ [73] | ViT-B | 86M | 800 | 4.1 | 3307 | 100 | 83.8 |
| SimMIM$_{192}$ [73] | Swin-B | 88M | 800 | 2.0 | 1609 | 100 | 84.0 |
| SimMIM$_{192}$ [73] | Swin-L | 197M | 800 | 3.5 | 2821 | 100 | 85.4 |
| **Ours** | **Swin-B** | **88M** | **800** | **1.1** | **887** | **100** | **83.8** |
| **Ours** | **Twins-L** | **99M** | **800** | **0.8** | **676** | **100** | **83.9** |
| **Ours** | **Swin-L** | **197M** | **800** | **1.3** | **1067** | **100** | **85.1** |

quadratic complexity of self-attention w.r.t. the window size. Fortunately, operating only on the visible patches permits the training with a larger window size with little extra cost. As displayed in Table 1, pre-training with doubled window size only marginally increases the training time/GPU memory by less than 10%/20% yet brings a moderate performance improvement despite $p = 7$ in the fine-tuning stage.

Table 1: A larger window size $p \times p$.

| $p \times p$ | Time | Mem. | Acc. |
|---|---|---|---|
| $7 \times 7$ | 1.1h | 121.6G | 83.2% |
| $14 \times 14$ | 1.2h | 148.9G | 83.4% |

## 4.3 ImageNet-1K Classification

We fine-tune the pre-trained models on the ImageNet-1K dataset and report the results on the validation set in Table 2. Here, we make direct comparisons with the models 1) trained from scratch with a longer training schedule, 2) trained with contrastive learning, and 3) trained with other MIM approaches. Our approach achieves 83.8%/83.9%/85.1% top-1 fine-tuning accuracy with the Swin-B/Twins-L/Swin-L backbone, which is clearly superior to the supervised learning/contrastive learning methods and on par with other MIM methods using backbones with similar capacities. The results demonstrate the effectiveness of our method, in addition to the substantial efficiency improvements over both MAE and SimMIM.

## 4.4 MS-COCO Object Detection and Instance Segmentation

Finally, we evaluate the transfer learning performance of our pre-trained models to the MS-COCO object detection and instance segmentation dataset. Here, we directly use the code base of the supervised Swin Transformer without any modification to the fine-tuning strategy. For direct comparisons, we reran the experiments for the supervised Swin-B and SimMIM using their public checkpoints. The experiment results are summarized in Table 3. Compared with the supervised pre-trained Swin-B, our approach performs prominently better in terms of all metrics, *e.g.*, 1.5% absolute improvement in $AP^b$. In addition, we also observed that our approach still performs comparably to the SimMIM on dense prediction tasks. More significantly, our approach outperforms most of the baselines from [45] using $3\times$ or $10\times$ more fine-tuning epochs and advanced data augmentations [20]. These experiments,

Table 3: **MS-COCO object detection and instance segmentation.** All methods are based on the Mask R-CNN [30] architecture with the FPN [46] neck. The methods in gray are cited from [45]. Most of them use much longer training schedules and advanced data augmentations.

| Method | Backbone | PT Ep. | PT Hours | FT Ep. | $AP^b$ | $AP^b_{50}$ | $AP^b_{75}$ | $AP^m$ | $AP^m_{50}$ | $AP^m_{75}$ |
|---|---|---|---|---|---|---|---|---|---|---|
| *Training from scratch* | | | | | | | | | | |
| Benchmarking [45] | ViT-B | 0 | 0 | 400 | 48.9 | - | - | 43.6 | - | - |
| *Supervised Pretraining* | | | | | | | | | | |
| Benchmarking [45] | ViT-B | 300 | 992 | 100 | 47.9 | - | - | 42.9 | - | - |
| PVT [66] | PVT-L | 300 | - | 36 | 44.5 | 66.0 | 48.3 | 40.7 | 63.4 | 43.7 |
| Swin [49] | Swin-B | 300 | 840 | 36 | 48.5 | 69.8 | 53.2 | 43.2 | 66.9 | 46.7 |
| *Self-Supervised Pre-training* | | | | | | | | | | |
| MoCov3 [12] | ViT-B | 800 | - | 100 | 47.9 | - | - | 42.7 | - | - |
| BEiT [3] | ViT-B | 800 | - | 100 | 49.8 | - | - | 44.4 | - | - |
| MAE [28] | ViT-B | 1600 | 2069 | 25 | 48.1 | - | - | - | - | - |
| MAE [28] | ViT-B | 1600 | 2069 | 100 | 50.3 | - | - | 44.9 | - | - |
| SimMIM [73] | Swin-B | 800 | 1609 | 36 | 50.4 | 70.9 | 55.5 | 44.4 | 68.2 | 47.9 |
| **Ours** | **Swin-B** | **800** | **887** | **36** | **50.0** | **70.7** | **55.4** | **44.1** | **67.9** | **47.5** |

combined with the results in Table 2, verify that our approach can achieve outstanding performance with impressive pre-training efficiency.

## 5 Conclusion

In this paper, we present a green approach for Masked Image Modeling (MIM) with hierarchical Vision Transformers, *e.g.*, Swin Transformer [49] and Twins Transformer [14], allowing the hierarchical models to discard masked patches and operate only on the visible ones. Coupling the efficient Group Window Attention scheme, the DP-algorithm-based Optimal Grouping strategy, and the Sparse Convolution, our approach can train the hierarchical models $\sim 2.7\times$ faster and reduce the GPU memory consumption by up to 70%, while still enjoying a competitive performance on ImageNet classification and the superiority of downstream MS-COCO object detection benchmarks. We hope that this work will facilitate future self-supervised learning methods targeting effectiveness as well as efficiency.

**Limitations.** One of the limitations of our algorithm is that it requires the batch-wise masking scheme (as in Section 3.5) to achieve the best efficiency. Although this limitation only has little impact on the MIM pre-training, it restrains the application of our method on a broader range of settings, e.g., training ViTs with token sparification [59, 75] that requires instance-wise sparsification. These applications are beyond the scope of this work and we will leave them for future study.

**Broader Impact.** This work proposed a green approach for MIM with hierarchical ViTs, prominently alleviating the heavy computation burden of MIM. On the one hand, this work provokes the efficiency as well as the effectiveness of MIM, which may inspire new algorithms and investigations in this direction. On the other hand, since the pre-training datasets might contain biases, our approach, in the same way as other unsupervised/self-supervised learning methods, might also be susceptible to replicating these biases. This concern can be mitigated by combining the FairML methods [4].

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
