Table 4: Influence of the optimal grouping on Group Window Attention.

| Group size $g_s$ | Dynamic Programming | FLOPS@Stage1 | FLOPS@Stage1 | FLOPS@Stage1 |
|---|---|---|---|---|
| Greedy (Algorithm 1) | ✓ | 62.6M | 55.4M | 52.3M |
| $p \times p$ (i.e., 49) | ✓ | 65.0M | 58.5M | 52.7M |
| $\max_i w_i$ | ✓ | 64.5M | 59.7M | 62.5M |
| $\max_i w_i$ | × | 137.5M | 113.3M | 75.7M |
| $\sum_i w_i$ | × | 201.3M | 69.2M | 52.7M |

Table 5: Time cost of each component in the Group Window Attention.

| Time Cost (ms) | DP | Masking | Pre-proc | Grouping | Ungrouping | Attention Fwd & Bwd |
|---|---|---|---|---|---|---|
| Stage1 | 4.9 | 3.8 | 14.8 | 0.4 | 0.4 | 61.9 |
| Stage2 | 0.6 | 0.1 | 2.1 | 0.2 | 0.2 | 20.8 |
| Stage3 | 0.1 | 0.1 | 1.0 | 0.1 | 0.1 | 15.4 |

# A   More Ablation Studies

## A.1   Efficiency comparison using Twins-L

Here we compare our method with a variant of MAE that operates on both visible and masked patches, which is almost identical to SimMIM except for the choice of the loss function. As shown in Table 6, our method performs on par with the baseline MAE operating on all patches while enjoying 2.6x pre-training speedup in a greener way.

Table 6: Comparison using Twins-L.

| Method | Time | Mem. | Acc. |
|---|---|---|---|
| MAE w/ all patches | 2.6h | 408.3G | 83.5% |
| Ours | 0.8h | 102.3G | 83.3% |

## A.2   Analysis on the Optimal Grouping

Following the setting in Figure 4, we compare the complexity of a single group attention module, w/ or w/o dynamic programming, at each stage as below. As displayed in Table 4, when $g_s = \max_i w_i$, the complexity is doubled without the DP solver. Simply setting $g_s = \sum_i w_i$ (such that there is only 1 group) also suffers from heavy cost, encountering an out-of-memory error in practice even with a much smaller batch size (e.g., 64 per GPU). In contrast, with the DP solver, the complexity is significantly reduced even when we simply fix the group size to the same as the window size $p \times p$ (note that $\max_i w_i = p \times p$ with a high probability in practice) as discussed in Sec. 4.2. This experiment demonstrates the efficacy of our optimal grouping scheme.

In addition, We benchmark the time cost (ms) of each component in a single Group Attention module and summarize the results in Table 5. Here we use a tensor of shape $[256, H_i \times W_i, C_i]$ as input, where $H_i, W_i, C_i$ denote the height, width, and the number of channels of the features in the $i$th stage. Note that the DP, masking, and other pre-processing operations are only executed twice for each stage, i.e., for the shifted/unshifted window partition. We can see that the extra cost of our method is indeed moderate compared with the attention computation.

# B   More Details of Our Method

## B.1   Method Overview.

We provide a diagram in Figure 6 for an intuitive illustration of our method. The input image is first randomly masked according to the masking scheme in Section 3.5, then fed into our green hierarchical ViT to obtain representations for each visible patch. Finally, we concatenate the representations of visible patches with the mask tokens and feed them into a transformer decoder, yielding the representations for the masked patches. Finally, we predict the raw pixel values of masked patches

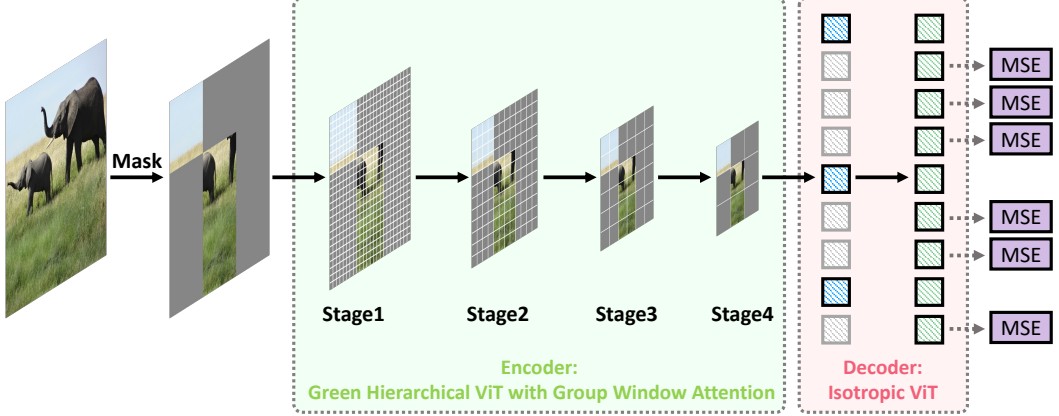

Figure 6: **Overview of our method.** The input image is randomly masked and then fed into a 4-stage hierarchical ViTs. Finally, a lightweight decoder takes the representations of visible patches and mask tokens to reconstruct the missing patches.

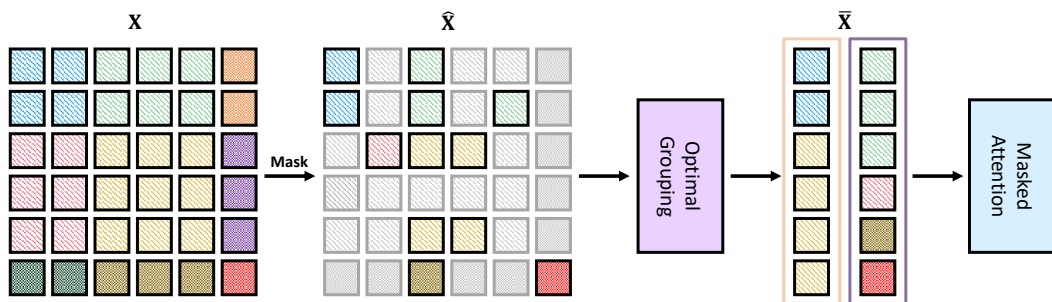

Figure 7: **Illustration of the Group Window Attention scheme with shifted windows.** It shows that our approach is agnostic to the window partition.

from the corresponding representations and implement the training by minimizing the Mean Square Error (MSE) between the predictions and ground truth.

## B.2 Group Window Attention scheme with shifted windows.

In addition to Figure 2, we also illustrate how our method works with the irregular window partition in Figure 7. We can observe that owing to the optimal grouping scheme, our method dynamically finds out the best group partition despite the number of visible patches within each local window being highly uneven. This figure further demonstrates that our approach is agnostic to the window partition and works impressively well.

## B.3 A Python Implementation of the Optimal Grouping algorithm

We provide a Python implementation of the Dynamic-Programming-based Optimal Grouping algorithm in Algorithm 2. As we can see, the two components of the Optimal Grouping algorithm are both easy to implement. For the DP algorithm for the single Knapsack problem, its time/space complexity is $\mathcal{O}(g_s n_w)$ where $g_s$ is the group size and $n_w$ is the number of windows. In practice, because $g_s$ and $n_w$ are generally small (*i.e.*, smaller than 100) the running time of Algorithm 2 is negligible (*i.e.*, <1ms).

## B.4 A PyTorch Implementation of the Group Attention scheme

With the group partition on the indexes, we can then permute visible patches according to the partition and obtain several groups of patches with an equal size $g_s$, upon which the Masked Attention is

**Algorithm 2** Dynamic Programming-based algorithm for the Optimal Grouping using Python.

```python
1  def Knapsack(g_s, Phi):
2      # g_s (int): Group size
3      # Phi (list[int]): The numbers of visible patches within each local window
4
5      n_w = len(Phi) # the number of windows
6      K = [[0 for w in range(g_s + 1)] for i in range(n_w + 1)] # a buffer for the DP
            algorithm
7
8      # Build table K[][] in a bottom up manner
9      for i in range(n_w + 1):
10         for w in range(g_s + 1):
11             if i == 0 or w == 0:
12                 K[i][w] = 0
13             elif Phi[i - 1] <= w:
14                 K[i][w] = max(Phi[i - 1] + K[i - 1][w - Phi[i - 1]], K[i - 1][w])
15             else:
16                 K[i][w] = K[i - 1][w]
17
18     # Store the result of Knapsack
19     res = K[n_w][g_s]
20
21     # Store the selected indexes
22     w = g_s
23     Pi = []
24
25     for i in range(n_w, 0, -1):
26         if res <= 0:
27             break
28
29         if res == K[i - 1][w]: # This window is not included.
30             continue
31         else: # This window is included.
32             Pi.append(i - 1)
33             # Since this window is included, its value is deducted
34             res = res - Phi[i - 1]
35             w = w - Phi[i - 1]
36
37     return Pi[::-1] # Optional: make Pi in an increasing order
38
39
40 def GroupPartition(g_s, Phi):
41     # g_s (int): Group size
42     # Phi (list[int]): The numbers of visible patches within each local window
43
44     win_szs = Phi.copy()
45     ori_win_idxs = list(range(len(win_szs)))
46     win_idxs = []
47
48     while len(win_szs) > 0:
49         idx = knapsack(group_size, win_szs)
50
51         # Append the selected idx
52         win_idxs.append([ori_win_idxs[i] for i in idx])
53
54         # The remaining windows and indexes
55         win_szs = [win_szs[i] for i in range(len(ori_win_idxs)) if i not in idx]
56         ori_win_idxs = [ori_win_idxs[i] for i in range(len(ori_win_idxs)) if i not in idx]
57
58     return win_idxs
```

performed. We also provide a PyTorch implementation of the Group Attention scheme in Algorithm 3 to facilitate future research. Note the padding operations are omitted here for simplicity.

**Algorithm 3** Group Attention using the PyTorch framework.

```python
1 def GroupAttention(x, g_s, Phi):
2     # x (3-d tensor): Features the visible patches
3     # g_s (int): Group size
4     # Phi (list[int]): The numbers of visible patches within each local window
5
6     # B is the batch size, L is the number of visible patches, C is the number of channels
7     B, L, C = x.shape
8
9     # Prepare for the group attention
10    win_idxs = GroupPartition(g_s, Phi)
11    patch_idxs = torch.arrange(sum(Phi))
12    patch_idxs = torch.split(patch_idxs, Phi)
13    shuffle_idxs = torch.cat([patch_idxs[wi] for wi in win_idxs])
14    unshuffle_idxs = torch.argsort(shuffle_idxs)
15
16    # Group partition. For simplicity, assume that the partition is even
17    x = torch.index_select(x, 1, shuffle_idxs) # (B, n_g * g_s, C)
18    x = x.reshape(-1, g_s, C) # (B * n_g, g_s, C)
19
20    # Attention with relative position bias as in Figure 3
21    x = MaskedAttention(x)
22
23    # Reverse the group partition
24    x = x.reshape(B, L, C).index_select(x, 1, unshuffle_idxs)
25
26    return x
```