# OpenReview forum: "Green Hierarchical Vision Transformer for Masked Image Modeling"
_NeurIPS.cc/2022/Conference — NeurIPS 2022 Accept_

### Official Review · Reviewer_Yf6s · 2022-07-06

**Rating:** 7
**Confidence:** 5
**Soundness:** 3 good
**Presentation:** 3 good
**Contribution:** 3 good

**Summary:**

In this paper, the authors propose a method to perform MAE style pre-training on Swin transformers. The main idea is to dynamically put the visible patches into groups and then perform attention within each group. To ensure that information doesn't "leak" out of each window, attention masking is performed in each group. This method can be efficient because of the Swin structure -- attention can only happen within a window, so the size of each group can be small and well-packed. The grouping is solved by dynamic programming to optimize FLOPs. The authors evaluate this approach on ImageNet and COCO and show that this method is efficient while being competitive in accuracy.

**Questions:**

1. I wonder if the authors have any intuition on why the accuracy of SimMIM is slightly higher than the proposed method.
2. I wonder if MIM on Swin works well with a larger backbone (e.g., Swin-L, Swin-H, etc.). I know that this might not be the main focus of this paper, but if MIM doesn't look promising on Swin, then speeding this up is less important. Conversely, if MIM on large Swin models works very well, then the importance of this work will be large.

**Limitations:**

I think the main limitation is that the method focuses on Swin Transformers only, but not other hierarchical models.

**Strengths And Weaknesses:**

Strengths:
+ The 2x speedup looks promising and useful. Given the wide use of Swin transformers, an efficient way to pre-train with SSL can be quite impactful.
+ The idea to perform grouping for fast MAE on Swin is interesting and novel.
+ The idea to optimize grouping based on FLOPs with DP is simple but seems to work well.
+ The accuracy on COCO suggests MIM+Swin is likely a good idea.


Weaknesses:
- The accuracy on ImageNet is a bit disappointing: Swin+MIM trained for 900 epochs obtains 83.7, while Swin-from-scratch trained for 300 epochs already obtains 83.5. Looking at ImageNet results alone, it's not clear if we need MIM on Swin.
- The title and writing overall suggests that the method works on general hierarchical ViT models, but in reality, the authors only show that the method works on Swin transformers. It's not just lack of experiments, it's not even clear how one can apply this approach to other hierarchical models (e.g., MViT).
- The experiments are limited to <=800 epoch training. Given that MIM is usually more useful in long-training regime, missing >800 epoch experiments make it harder to compare to MAE.
- How much time does the DP solver take? How much time does the grouping, masking, or ungrouping take? Some more detailed analysis would be helpful.
- (minor:) In figure 1, without accuracy, it's hard to appreciate the method by speed only.

---

> ### Author Response · Authors · 2022-08-02
> **Response To Reviewer Yf6s (1/2)**
>
> We thank the reviewer for the valuable comments and respond to them appropriately as follows. We will add suggested experiments and explanations in the updated manuscript.
>
> > Q1. The accuracy on ImageNet is a bit disappointing: Swin+MIM trained for 900 epochs obtains 83.7, while Swin-from-scratch trained for 300 epochs already obtains 83.5. Looking at ImageNet results alone, it's not clear if we need MIM on Swin.
>
> A1. Even if we only look at the Imagenet results, we believe that the performance of MIM with Swin is promising compared with the supervised baseline:
> 1. **Longer training schedule does NOT further improve the supervised baseline.** As shown in Table 7 of SimMIM [1], Swin-B with 800 epochs of supervised pretraining and 100 epochs of finetuning achieves only 83.3% accuracy on ImageNet. In contrast, our method achieves 83.8% accuracy under the same setting.
> 2. **Our method performs much better than the supervised baseline when using a larger model.** With the Swin-L, our method backbone achieves 85.1% accuracy (see A7 for more details) while the supervised baseline only obtains 83.5% accuracy (as in Table 7 of SimMIM [1]).
>
>
> > Q2. Generality of the proposed method.
>
> A2. Thanks for the suggestion. While we chose the representative _pure_ hierarchical ViT--Swin Transformers--as our standpoint, our method can easily generalize to other _hybrid_ hierarchical ViTs (e.g., ViTs with convolution or pooling layers) with minimal adaptions. Our intuition is that the features of visible patches can be viewed as sparse tensors, upon which the convolution/pooling operation can be performed efficiently with the help of Sparse Convolution [1] (implemented for newer GPUs in [2,3]), originally designed for 3D point cloud data. Here we take the Twins Transformers [4], which contain both window attention and (depthwise) convolution, as a concrete example. We apply our method to the Twins-L model (98M parameters, similar to Swin-B) and replace all the standard convolutions with sparse convolutions. Due to the time limitation of the response period, we are only able to provide the results of 100 pre-training epochs. Yet, we can still observe that our method performs on par with the baseline MAE operating on all patches while enjoying 2.6x pre-training speedup in a greener way.
>
> | Method | PT Resolution | GPU Hours | GPU Memory | IN-1K Acc. |
> |:-------|:-------------:|:---------:|:----------:|:---------:|
> | MAE with ALL patches | 224x224 | 261.3 | 408.3GB | 83.5% |
> |Ours for Twins   | 224x224 | 84.6 | 102.3GB | 83.3% |
>
> We will clarify this part, articulate how our method can generalize to most hierarchical ViTs, and include more experiment results of longer training schedules with various ViTs in the updated manuscript. In addition, to facilitate future research, we will release all the pre-trained models, relevant code, and detailed instructions to pre-train customized hierarchical ViTs with our method.
>
> > Q3. Pretraining with longer training epochs.
>
> A3. We conduct experiment to evaluate our method with longer training epochs and summarize the result in the following table, where we can see that our method with 1600 epochs achieve 83.9% accuracy. Note that, due to the time limitation, we directly use the same parameters as 800-epochs pre-training/fine-tuning, which may be suboptimal compared with SimMIM using optimized hyper-parameters for different schedules. We may expect a further performance boost with better training configurations.
>
> | Method | PT Epochs | GPU Hours | GPU Memory | IN-1K Acc. |
> |:-------|:-------------:|:---------:|:----------:|:---------:|
> | SimMIM$_{192}$ | 800 | 1609 | 284.8GB | 84.0% |
> | SimMIM$_{224}$ | 800 | 2251 | 398.3GB | 84.1% |
> | Ours   | 800 | 887  | 121.6GB | 83.8% |
> | Ours   | 1600| 1774 | 121.6GB | 83.9% |
>
> *Note that the fine-tuning performance of our model is slightly boosted by 0.1% compared with the reported number in the paper, following SimMIM to use a fine-tuning batch size of 2048.
>
> [1] SimMIM: a Simple Framework for Masked Image Modeling. Xie et al., CVPR2022.
>
> [2] Graham et al., _3D Semantic Segmentation with Submanifold Sparse Convolutional Networks_. CVPR'2018.
>
> [3] Choy et al., _4D Spatio-Temporal ConvNets: Minkowski Convolutional Neural Networks_. CVPR'2019.
>
> [4] https://github.com/traveller59/spconv (Apache-2.0 License).
>
> [5] Chu et al., _Twins: Revisiting the Design of Spatial Attention in Vision Transformers_. NeurIPS'2021.

---

> ### Author Response · Authors · 2022-08-02
> **Response To Reviewer Yf6s (2/2)**
>
> > Q4. How much time does the DP solver take? How much time does the grouping, masking, or ungrouping take? Some more detailed analysis would be helpful.
>
> A4. Thanks for the comment. We benchmark the time cost (ms) of each component in a single Group Attention module and summarize the results in the table below.
>
> |Time Cost (ms)| DP | Masking | Pre-proc | Grouping | Ungrouping | Attention Fwd&Bwd |
> |:-------------|:---------:|:-------:|:--------------:|:--------:|:----------:|:---------:|
> | Stage1 | 4.9 | 3.8 | 14.8 | 0.4 | 0.4 | 61.9 |
> | Stage2 | 0.6 | 0.1 | 2.1  | 0.2 | 0.2 | 20.8 |
> | Stage3 | 0.1 | 0.1 | 1.0  | 0.1 | 0.1 | 15.4 |
>
> Here we use a tensor of shape $[256, H_i\times W_i, C_i]$ as input, where $H_i, W_i, C_i$ denote the height, width, and the number of channels of the features in the $i$th stage. Note that the DP, masking, and other pre-processing operations are only executed twice for each stage, i.e., for the shifted/unshifted window partition. We can see that the extra cost of our method is indeed moderate compared with the attention computation.
>
>
> > Q5. Add an accuracy comparison to Fig. 1.
>
> A5. We will follow your suggestion to add another subfigure of accuracy in Fig. 1 for a more intuitive comparison.
>
> > Q6. Why SimMIM is slightly better than the proposed method.
>
> A6. There might be multiple explanations for this question. First, as in the eighth and ninth rows of Table 2 in our manuscript, we can see that SimMIM also slightly outperforms MAE (which is our standpoint) when using the same ViT-Base backbone. Second, unlike the isotropic ViTs that exchange information between all visible patches, the window-based Swin Transformer can only process a local window of patches in a time and, thereby, the intermediate features of the masked patches may act as proxies for information propogation.
>
> > Q7. Scaling to larger models.
>
> A7. Actually, in Appendix A of our supplementary material, we evaluated our method on the larger Swin-L backbone, which has $2\times$ parameters compared with the Swin-B. The results are summarized below. We can observe that our method still obtains competitive performance on IN-1K with even larger efficiency improvements, i.e., $\sim 2.7\times$ speedup and only $\sim 30$% of GPU memory consumption, highlighting the efficacy of our method. We will put this result into the main body of our paper to emphasize the scaling ability of our method.
>
> | Method | PT Resolution | GPU Hours | GPU Memory | IN-1K Acc. |
> |:-------|:-------------:|:---------:|:------------:|:-----------------:|
> | SimMIM | 192x192 | 2821 | 727.0GB | 85.4 |
> | Ours   | 224x224 | 1067 | 233.6GB | 85.1 |

---

> ### Author Response · Authors · 2022-08-09
> **Looking Forward to Hearing from Reviewer Yf6s**
>
> Dear reviewer, thanks again for the careful reviews and constructive suggestions. We have provided additional explanations and experiments to address the concerns accordingly. Since the deadline of the author-reviewer discussion period is approaching soon, we would like to discuss with you whether or not the concerns have been resolved. And if you have any additional comments, we are happy to answer them further.

---

### Official Review · Reviewer_dAsW · 2022-07-11

**Rating:** 5
**Confidence:** 4
**Soundness:** 3 good
**Presentation:** 3 good
**Contribution:** 3 good

**Summary:**

The paper first introduces to properly apply the MAE pre-training method on Swin Transformer backbone architecture. To adapt the visible-only encoder with local window attention in Swin, the paper proposes a new Group Window attention mechanism with Dynamic Programming to automatically tune the group size and local windows. The introduced method significantly reduces the pre-training hours of SimMIM on Swin Transformer with marginal performance drops.

**Questions:**

I like the motivation and story of this work but would like to see more experiments that would make it more solid. Please see Weaknesses for details.

**Limitations:**

The authors discussed the limitation of the batch-wise masking scheme in the Appendix.

**Strengths And Weaknesses:**

Strengths:
+ Writing: The paper is generally well written and easy to follow. The story of green AI is interesting.
+ Contribution: It is significant to well apply the MAE-style pretraining method on hierarchical Vision Transformer architectures, especially benefitting dense prediction downstream tasks, e.g., detection and segmentation. It is the first work to investigate this problem so far as I know.

Weaknesses:

I am generally satisfied with this work but have several concerns about the experiments:
1. **Marginal improvements:** The improvements over training from scratch on ImageNet is marginal, i.e., 83.5 vs. 83.7. The pretraining hours are comparable (840 for training from scratch and 887 for this work), it may raise a question that if 83.7 can also be achieved by training from scratch for 887 hours. By the way, why not compare to SimMIM_224 on Swin-Base?
2. **Lack of ablation studies:** (i) I think it is important to implement this method on the vanilla MAE with ViT-Base to show the performance gaps caused by group window attention. (ii) I would like to see if the performance can be further improved with more training epochs. For example, for 800 epochs, SimMIM_192 takes double the training hours than this work but +0.3% performance gains. I want to know if this method can also achieve 84.0% with double training hours (800ep -> 1600ep).
3. **More model scales:** It is better to show the power of this method on other sizes of Swin Transformer, e.g., small, large, etc.

---

> ### Author Response · Authors · 2022-08-02
> **Response To Reviewer dAsW**
>
> We thank the reviewer for the valuable comments and respond to them appropriately as follows. We will add suggested experiments and explanations in the updated manuscript.
>
> > Q1. Performance improvement.
>
> A1. We respectfully disagree that the performance improvement is marginal over the supervised baseline:
> 1. **Longer training schedule does NOT necessarily improve the supervised baseline.** As shown in Table 7 of SimMIM [1], Swin-B with 800 epochs of supervised pretraining and 100 epochs of finetuning achieves only 83.3% accuracy on ImageNet.
> 2. **Our method performs much better than the supervised baseline when using a larger model.** With the Swin-L, our method achieves 85.1% accuracy (see A5 for more details) while the supervised baseline only obtains 83.5% accuracy (as in Table 7 of SimMIM [1]).
> 3. **Our method performs much better on transfer learning than the supervised baseline.** As displayed in Table 3 of our paper, when transferred to MS-COCO object detection, the model pre-trained with our method substantially outperforms the one with supervised pre-trained by 1.6% mIoU.
>
> > Q2. Why not compare with SimMIM$_{224}$?
>
> A2. We were unable to directly compare with SimMIM_{224} in terms of performance because the original paper did not provide any results of SimMIM_{224} with Swin Transformer. Here we train a Swin-B model with SimMIM_{224} with the official code and make comparisons with our method in the following table. We can see that SimMIM_{224} performs similarly to our method, but it invloves a sharp increase on training cost.
>
> | Method | PT Resolution | GPU Hours | GPU Memory | IN-1K Acc. |
> |:-------|:-------------:|:---------:|:----------:|:---------:|
> | SimMIM | 192x192 | 1609 | 284.8GB | 84.0% |
> | SimMIM | 224x224 | 2251 | 398.3GB | 84.1% |
> | Ours   | 224x224 | 887  | 121.6GB | 83.8%* |
>
> *Note that the fine-tuning performance of our model is slightly boosted by 0.1% compared with the reported number in the paper, following SimMIM to use a fine-tuning batch size of 2048.
>
> > Q3. I think it is important to implement this method on the vanilla MAE with ViT-Base to show the performance gaps caused by group window attention.
>
> A3. We note that the proposed method would be the same as MAE when using the isotropic ViT-Base model. The motivation of our method is to alleviate the drawback of MAE (e.g. weak performance for dense predictions) and generalize it to hierarchical ViTs.
>
> > Q4. Pre-training with longer training epochs.
>
> A4. We conduct experiment to evaluate our method with longer training epochs and summarize the result in the following table, where we can see that our method with 1600 epochs achieve 83.9% accuracy. Note that, due to the time limitation, we directly use the same parameters as 800-epochs pre-training/fine-tuning, which may be suboptimal compared with SimMIM using optimized hyper-parameters for different schedules. We may expect a further performance boost with better training configurations.
>
> | Method | PT Epochs | GPU Hours | GPU Memory | IN-1K Acc. |
> |:-------|:-------------:|:---------:|:----------:|:---------:|
> | SimMIM$_{192}$ | 800 | 1609 | 284.8GB | 84.0% |
> | SimMIM$_{224}$ | 800 | 2251 | 398.3GB | 84.1% |
> | Ours   | 800 | 887  | 121.6GB | 83.8% |
> | Ours   | 1600| 1774 | 121.6GB | 83.9% |
>
>
> > Q5. Scaling to other model size.
>
> A5. We actually have evaluated our method on the larger Swin-L backbone in supplementary materials. The results are summarized below (and also in Table 4 of our supplementary material). We can observe that our method still obtains competitive performance on IN-1K with even larger efficiency improvements, i.e., $\sim 2.7\times$ speedup and only $\sim 30$% of GPU memory consumption, highlighting the efficacy of our method.
>
> | Method | PT Resolution | GPU Hours | GPU Memory | IN-1K Acc. |
> |:-------|:-------------:|:---------:|:------------:|:-----------------:|
> | SimMIM | 192x192 | 2821 | 727.0GB | 85.4% |
> | Ours   | 224x224 | 1067 | 233.6GB | 85.1% |
>
>
>
> [1] SimMIM: a Simple Framework for Masked Image Modeling. Xie et al., CVPR2022.

---

> > ### Comment · Reviewer_dAsW · 2022-08-04
> > **Further Questions**
> >
> > Thank you for the response. Regarding Q3, I am wondering if your method could outperform the original MAE on top of ViT backbones, that is, replacing MAE's vanilla random masking with your group window masking mechanism (Fig. 2).

---

> > > ### Author Response · Authors · 2022-08-05
> > > **Regarding the Group Window Attention in Fig. 2**
> > >
> > > Thanks for the question. The reviewer might have some misunderstandings about our Group Window Attention in Fig. 2, which aims to address the incapability of the local window attention in hierarchical vision transformers to handle the windows of arbitrary size caused by the random masking. We would like to clarify that, because the isotropic ViT backbones use global attention in all of their building blocks, window grouping is not necessary anymore since there is only ONE window. Therefore, our method will perform exactly the same as MAE if we use the isotropic ViT backbones.

---

> > > > ### Comment · Reviewer_dAsW · 2022-08-07
> > > > **Thank you for the response**
> > > >
> > > > Thanks, I have no more questions.

---

### Official Review · Reviewer_RDo2 · 2022-07-11

**Rating:** 5
**Confidence:** 5
**Soundness:** 3 good
**Presentation:** 2 fair
**Contribution:** 2 fair

**Summary:**

This paper presents a cost-effective approach for masked image modeling (MIM) pretraining with hierarchical vision transformers. MIM has been proved effective for vision transformer pretraining, but hierarchical ViT faces a challenge in MIM pretraining that both visible and masked patches have to be involved, which greatly decreases the pretraining efficiency. This paper presents a solution which reduces the pretraining cost of Swin Transformer by around half while achieving on par performance.

**Questions:**

Please refer to the previous section.

**Limitations:**

No potential negative societal impact is identified.

**Strengths And Weaknesses:**

This paper addresses a relevant problem with a technically sound solution. The cost of MIM pretraining for hierarchical ViTs is indeed a painpoint in vision backbone learning. The proposed group window attention scheme is reasonable, and the evaluation on Swin Transformer is c.

The paper claims that the proposed approach is for hierarchical ViTs, but the experiments are carried out only for Swin Transformer. It would be more convincing if the authors could show the results for at least another ViT. Or, the authors should consider not to claim the proposed approach to be generally applicable to hierarchical ViTs.

The group window attention algorithm solves an optimization problem to determine the group size and the number of groups. The sole optimization target is the computation cost, leaving the pretrain quality completely unconsidered. Does the selection of group size affect the pretrain quality?

The authors use dynamic programming to solve the group partition problem, but does a simple and heuristic algorithm, such as g_s=max{w_i}, already suffices? The complexity of the dynamic programming algorithm is not well justified/ablated.

---

> ### Author Response · Authors · 2022-08-02
> **Response To Reviewer RDo2**
>
> We thank the reviewer for the valuable comments and respond to them appropriately as follows. We will add suggested experiments and explanations in the updated manuscript.
>
> > Q1. The paper claims that the proposed approach is for hierarchical ViTs, but the experiments are carried out only for Swin Transformer. It would be more convincing if the authors could show the results for at least another ViT. Or, the authors should consider not to claim the proposed approach to be generally applicable to hierarchical ViTs.
>
> A1. Thanks for the suggestion. While we chose the representative _pure_ hierarchical ViT--Swin Transformers--as our standpoint, our method can easily generalize to other _hybrid_ hierarchical ViTs (e.g., ViTs with convolution or pooling layers) with minimal adaptions. Our intuition is that the features of visible patches can be viewed as sparse tensors, upon which the convolution/pooling operation can be performed efficiently with the help of Sparse Convolution [1] (implemented for the newer GPUs in [2,3]), originally designed for 3D point cloud data. Here we take the Twins Transformers [4], which contain both window attention and (depthwise) convolution, as a concrete example. We apply our method to the Twins-L model (98M parameters, similar to Swin-B) and replace all the standard convolutions with sparse convolutions. Due to the time limitation of the response period, we are only able to provide the results of 100 pre-training epochs. Yet, we can still observe that our method performs on par with the baseline MAE operating on all patches while enjoying 2.6x pre-training speedup in a greener way.
>
> | Method | PT Resolution | GPU Hours | GPU Memory | IN-1K Acc. |
> |:-------|:-------------:|:---------:|:----------:|:---------:|
> | MAE with ALL patches | 224x224 | 261.3 | 408.3GB | 83.5% |
> |Ours for Twins | 224x224 | 84.6 | 102.3GB | 83.3% |
>
> We will clarify this part, articulate how our method can generalize to the most hierarchical ViTs, and include more experiment results of longer training schedules for various ViTs in the updated manuscript. In addition, to facilitate future research, we will release all the pre-trained models, relevant code, and detailed instructions to pre-train customized hierarchical ViTs with our method.
>
>
> > Q2. The sole optimization target is the computation cost, leaving the pretrain quality completely unconsidered. Does the selection of group size affect the pretrain quality?
>
> A2. Because we adopt the Mask Attention scheme (as in Sec. 3.4 and Fig. 3) that ensures the tokens from different local windows have NO interaction, the choice of group size has no impact on the pretrain quality. As a result, we can focus only on optimizing the training efficiency.
>
> > Q3. The authors use dynamic programming to solve the group partition problem, but does a simple and heuristic algorithm, such as $g_s = \max~{w_i}$, already suffices? The complexity of the dynamic programming algorithm is not well justified/ablated.
>
> A3. Following the setting in Fig. 4, we compare the complexity of a single group attention module, w/ or w/o dynamic programming, at each stage as below. We can see that, when $g_s = \max_i w_i$, the complexity is doubled without the DP solver. Simply setting $g_s = \sum_i w_i$ (such that there is only 1 group) also suffers from heavy cost, encountering an out-of-memory error in practice even with a much smaller batch size (e.g., 64 per GPU). In contrast, with the DP solver, the complexity is significantly reduced even when we simply fix the group size to the same as the window size $p\times p$ (note that $\max_i w_i = p\times p$ with a high probability in practice) as discussed in L272 of our original submission. This experiment demonstrates the efficacy of our optimal grouping scheme.
>
> | Group size $g_s$| Dynamic Programming | FLOPs@Stage1 | FLOPs@Stage2 | FLOPs@Stage3 |
> |:-----------|:-------------------:|:------------:|:-----------------:|:------------:|
> | Greedy (Alg. 1)| $\checkmark$ | **62.6M** | **55.4M** | **52.3M** |
> | $p \times p$ (i.e., 49) | $\checkmark$ | 65.0M | 58.5M | 52.7M |
> | $\max_i w_i$| $\checkmark$ | 64.5M | 59.7M | 62.5M |
> | $\max_i w_i$ | $\times$ | 137.5M | 113.3M | 75.7M |
> | $\sum_i~w_i$ | $\times$ | 201.3M | 69.2M | 52.7M |
>
>
> [1] Graham et al., _3D Semantic Segmentation with Submanifold Sparse Convolutional Networks_. CVPR'2018.
>
> [2] Choy et al., _4D Spatio-Temporal ConvNets: Minkowski Convolutional Neural Networks_. CVPR'2019.
>
> [3] https://github.com/traveller59/spconv (Apache-2.0 License).
>
> [4] Chu et al., _Twins: Revisiting the Design of Spatial Attention in Vision Transformers_. NeurIPS'2021.

---

> ### Author Response · Authors · 2022-08-09
> **Looking Forward to Hearing from Reviewer RDo2**
>
> Dear reviewer, thanks again for the careful reviews and constructive suggestions. We have provided additional explanations and experiments to address the concerns accordingly. Since the deadline of the author-reviewer discussion period is approaching soon, we would like to discuss with you whether or not the concerns have been resolved. And if you have any additional comments, we are happy to answer them further.

---

> > ### Comment · Reviewer_RDo2 · 2022-08-10
> > **Thanks for the response.**
> >
> > I do not have further questions.

---

### Meta-Review · Area_Chair_6HoS · 2022-08-25

**Recommendation:** Accept
**Confidence:** Certain

**Metareview:**

After rebuttal and discussion all reviewers recommend acceptance. The AC sees no reason to overturn this recommendation.

**Award:**

No

---

### Decision · Program_Chairs · 2022-09-14

Accept